# Microbial Production of *N*-Acetylneuraminic Acid Using Metabolically Engineered *Escherichia coli* and *Bacillus subtilis*: Advances and Perspectives

**DOI:** 10.3390/foods14203478

**Published:** 2025-10-12

**Authors:** Jingru Dang, Zhijie Shi, Heyun Wu, Qian Ma, Xixian Xie

**Affiliations:** 1Key Laboratory of Industrial Fermentation Microbiology, Ministry of Education, Tianjin University of Science and Technology, Tianjin 300457, China; djr18031179958@163.com (J.D.); 18294197868@163.com (Z.S.); wuheyun@tust.edu.cn (H.W.); 2College of Biotechnology, Tianjin University of Science and Technology, Tianjin 300457, China; 3National and Local United Engineering Lab of Metabolic Control Fermentation Technology, Tianjin University of Science and Technology, Tianjin 300457, China

**Keywords:** Neu5Ac, cell factory, metabolic engineering, *Escherichia coli*, *Bacillus subtilis*

## Abstract

*N*-Acetylneuraminic acid (Neu5Ac), the predominant form of sialic acids (Sias), is extensively utilized in the food, pharmaceutical, and cosmetic industries. Microbial fermentation serves as a critical production method for its economical, eco-friendly, and scalable production. *Escherichia coli* and *Bacillus subtilis*, as primary industrial workhorses for Neu5Ac production, have been extensively investigated owing to their well-characterized genetic frameworks and mature molecular toolkits. Nevertheless, the intricate regulatory networks inherent to microbial systems present formidable obstacles to the high-efficiency biosynthesis of Neu5Ac. This review delineates the genetic and molecular mechanisms underlying Neu5Ac biosynthesis in both *E. coli* and *B. subtilis*. Furthermore, the rational and irrational strategies for constructing Neu5Ac microbial cell factories are systematically summarized, including the application of rational metabolic engineering to relieve feedback regulation, reconfigure metabolic networks, implement dynamic regulation, and optimize carbon sources; as well as the use of irrational strategies including directed evolution of key enzymes and high-throughput screening based on biosensors. Finally, this review addresses current challenges in Neu5Ac bioproduction and proposes integrative solutions combining machine learning with systems metabolic engineering to advance the construction of high-titer Neu5Ac microbial cell factory and the refinement of advanced fermentation technologies.

## 1. Introduction

*N*-Acetylneuraminic acid (Neu5Ac), an α-ketoacid sugar featuring a nine-carbon backbone [1], serves as the primary form of sialic acids (Sias). It is ubiquitously distributed among viruses, bacteria, protozoa, and higher organisms [2]. Endowed with diverse biological functions, including antiviral [3], antioxidant [4], anti-adhesive [5], anti-inflammatory [6], enhancement of brain function [7], facilitation of bone development [8], and skin-whitening effects [9], Neu5Ac is intricately involved in numerous crucial physiological and pathological processes [10]. Consequently, Neu5Ac holds substantial application potential across various domains such as functional foods, pharmaceuticals, and cosmetics (Figure 1). It can be employed in dietary supplements [11], serves as a pivotal precursor for the chemical synthesis of anti-influenza drugs (e.g., zanamivir and oseltamivir) [12], and is utilized in targeted cancer therapies [13]. According to market reports, pharmaceuticals, food, cosmetics and other areas account for approximately 46%, 29%, 15% and 10% of total market demand for Sias, respectively [14]. The global market of Sias reached $560 million in 2024 and is projected to grow at a 7.5% CAGR, reaching $1.08 billion by 2033 [14].

The efficient production of Neu5Ac has emerged as a focus of research, attributable to its extensive application prospects. In conventional preparation methods, Neu5Ac can be extracted and purified from natural raw materials such as milk, edible bird’s nests, and egg yolks. Nevertheless, the extraction process is intricate and exhibits a low recovery rate [15]; chemical synthesis of Neu5Ac necessitates stringent reaction conditions and costly toxic metals as catalysts, thereby resulting in low titers and substantial environmental pollution [16]; enzymatic synthesis of Neu5Ac demands the addition of the cofactor ATP during the catalytic process, and the purification steps of the enzyme are intricate and time-consuming [17]. To overcome these limitations and facilitate large-scale industrial production of Neu5Ac, the whole-cell catalysis approach has been developed and comprehensively investigated [18,19]. However, since this method utilizes high concentrations of *N*-acetylglucosamine (GlcNAc) and pyruvate as substrates, and there is a relatively large amount of substrate residue in the system, the cost of separation and extraction is high, thereby diminishing the economic competitiveness of this process [20]. Against this background, microbial fermentation has emerged as a promising alternative strategy. It has drawn significant attention owing to its numerous advantages, including high cost-effectiveness, operational simplicity, facile controllability, minimal by-product generation, high titer, and environmental benignity [21,22]. At present, several microorganisms have been genetically engineered for Neu5Ac production. Among them, *Escherichia coli* has distinguished itself by virtue of its rapid growth rate, high robustness, well-characterized genetic background, and efficient protein expression system [23]. However, high-density fermentation of *E. coli* still encounters a notable bottleneck: endotoxins are prone to be released into the fermentation broth, leading to product contamination [24]. Consequently, additional treatment is indispensable to ensure its safe application in the food and pharmaceutical industries. In contrast, *Bacillus subtilis*, as a Gram-positive model bacterium, is non-pathogenic, endotoxin-free, and has attained the Generally Recognized As Safe (GRAS) status [25,26]. These attributes endow it with potential utility in the production of food-grade Neu5Ac. Moreover, the Neu5Ac titer in *E. coli* has reached the highest reported level of 77.12 g/L [27], exceeding the 30.10 g/L achieved in *B. subtilis* [21]. This disparity may stem from differences in the growth and metabolism of these two host cells, *B. subtilis* exhibits a longer growth cycle compared to *E. coli*, probably resulting in a slower supply rate of the precursor PEP [28,29]. In comparison, other microorganisms are currently not the preferred choices as chassis strains for Neu5Ac production within the existing literature. Take *Corynebacterium glutamicum*, a common chassis strain for producing the Neu5Ac precursor GlcNAc [30], as an example. On one hand, it has a Neu5Ac catabolic pathway [31]; on the other, its relatively long growth cycle complicates the shift from the growth to the production phase, which makes it tough to balance the supply of GlcNAc and PEP. Moreover, natural Neu5Ac-producing strains like *E. coli* K1, *Campylobacter jejuni*, and *Neisseria meningitidis* are pathogenic, raising biosafety concerns. Therefore, this review aims to comprehensively summarize the production advances of Neu5Ac in *E. coli* and *B. subtilis*.

Over recent decades, the traditional strategy of utilizing *E. coli* or *B. subtilis* as chassis cells and constructing engineered strains via mutagenesis screening has been documented [32]. Although these conventional mutagenesis breeding strategies have attained a certain degree of success in screening Neu5Ac-producing strains with high titers, they are severely constrained in their industrial applications due to several drawbacks. These include a high level of randomness in mutagenesis, numerous uncertain factors, intricate screening procedures, and the necessity for continuous resistance screening using growth-limiting factors [33,34]. In recent years, the advent of metabolic engineering and synthetic biology has fundamentally transformed the realm of strain development. The development of technologies such as gene modification and metabolic flux optimization based on omics technologies [35], dynamic regulation based on N-terminal coding sequences (NCSs) and population quality control systems [36,37], high-throughput screening (HTS) based on biosensors [38], modular pathway engineering [21], riboswitch evolution engineering [39], and promoter engineering [40] has significantly expedited the development of high-titer Neu5Ac strains. Nevertheless, the equilibrium between cell growth and product synthesis remains a central concern in the efficient synthesis of Neu5Ac. Constructing high-performance Neu5Ac cell factories still presents a variety of challenges, such as the insufficiency of precursor supply, competition among branch pathways for metabolic flux, the existence of feedback inhibition and repression. Previous reviews on Neu5Ac mainly focused on their functional applications and the whole-cell catalytic synthesis. However, there is currently no comprehensive review discussing and analyzing recent advancements in the microbial synthesis of Neu5Ac via rational metabolic engineering strategies and irrational engineering strategies over the past three to five years.

This review comprehensively examines the synthetic pathways and regulatory mechanisms of Neu5Ac in *E. coli* and *B. subtilis*, along with its upstream production methods and downstream purification challenges. The entire metabolic pathway is partitioned into two modules: central metabolism and Neu5Ac biosynthesis, corresponding to the growth and production phases of fermentation. Based on this framework, the pivotal metabolic pathways of Neu5Ac are elucidated, along with the genetic regulatory mechanisms mediated by feedback regulation. Furthermore, it systematically summarized a range of rational and irrational strategies employed to enhance Neu5Ac production. The rational strategies include alleviating feedback regulation, reconstructing metabolic networks, implementing dynamic regulation strategies, and optimizing carbon sources. The irrational strategies include directed evolution of key enzymes and HTS based on biosensors. Finally, the challenges encountered in Neu5Ac production are critically discussed. To this end, a solution integrating machine learning and systems metabolic engineering is proposed, with the overarching aim of facilitating the construction of high-titer Neu5Ac strains and advancing the development of fermentation processes.

## 2. Biosynthetic Pathway, Regulation Mechanisms, and Production of Neu5Ac

### 2.1. Biosynthetic Pathway of Neu5Ac

The biosynthetic pathway of Neu5Ac can be delineated into a central metabolic module, which supplies precursors, and a Neu5Ac synthesis module. Specifically, the central metabolic module includes the glucose uptake system, glycolysis (EMP), the pentose phosphate pathway, the tricarboxylic acid (TCA) cycle, the glyoxylate pathway, as well as the biosynthesis and degradation of fatty acids [41,42]; meanwhile, the Neu5Ac synthesis module principally consists of the GlcNAc 2-epimerase (AGE) pathway, the GlcNAc-6-phosphate 2-epimerase (NanE) pathway, and the UDP-GlcNAc 2-epimerase (NeuC) pathway [21] (Figure 2).

Initially, glucose is taken up into the cell mainly through the phosphotransferase system (PTS), while glycerol enters via non-PTS, and is converted into glucose-6-phosphate (G6P) or glycerol-3-phosphate (G3P), respectively [43]. Subsequently, within the framework of the EMP pathway, a fraction of the metabolic flux is directed by citrate synthase towards the generation of acetyl-CoA, which then enters the TCA cycle. In *E. coli*, citrate synthase is encoded by *gltA* [44], and in *B. subtilis*, by *citZ* and *citA* [45]. Another portion of the metabolic flux proceeds through the anaplerotic pathway of oxaloacetate. Here, the phosphoenolpyruvate carboxylase encoded by *ppc* or the phosphoenolpyruvate carboxykinase encoded by *pck* is utilized to mediate the interconversion between phosphoenolpyruvate (PEP) and oxaloacetate [46,47], thereby replenishing the TCA cycle. These reactions play a pivotal role in the production and degradation of PEP, a key precursor for Neu5Ac synthesis. Furthermore, when the carbon source is in excess, *E. coli* and *B. subtilis* cannot fully activate the TCA cycle, which leads to the decomposition of pyruvate and the formation of by-products like acetic acid and lactic acid in *E. coli*, and acetoin in *B. subtilis* [48], resulting in a reduction in the metabolic flux directed towards the Neu5Ac synthesis pathway. In *E. coli*, the enzymes pyruvate oxidase, phosphotransacetylase, acetate kinase, and lactate dehydrogenase, which catalyze pyruvate conversion into acetic acid and lactic acid, are encoded by the genes *poxB*, *pta*, *ackA*, and *ldhA*, respectively, whereas in *B. subtilis*, acetolactate synthase and acetolactate decarboxylase, responsible for converting pyruvate into acetoin, are encoded by *alsSD* [49,50,51,52]. Notably, in *B. subtilis*, malic enzyme (YtsJ) can also convert malate to pyruvate [53], thereby supplementing the TCA cycle.

In microorganisms, the synthesis module of Neu5Ac commences when glucosamine-6-phosphate synthase (GlmS) catalyzes the conversion of fructose-6-phosphate (F6P) into glucosamine-6-phosphate (GlcN6P). This enzyme is subject to feedback inhibition by GlcN6P [54]. Following this, Neu5Ac is produced via three distinct pathways: (1) AGE pathway, with glucosamine-6-phosphate N-acetyl-transferase 1 (GNA1) and AGE as key enzymes; (2) NanE pathway, with GNA1 and NanE as key enzymes; and (3) NeuC pathway, with NeuC as the key enzyme. As the shared product of these three pathways, ManNAc can be further transformed into Neu5Ac either by the key enzyme NeuB, utilizing PEP as a co-substrate, or by the key enzyme NanA, utilizing pyruvate as a co-substrate. Obviously, the selection of the source of key enzymes is crucial to the synthesis efficiency of Neu5Ac. At present, *E. coli* W3110 NEA-27, achieving the highest Neu5Ac titer, employs key enzymes AGE from *Anabaena* sp. CH1 and NeuB from *N. meningitidis* [27]. Moreover, many other studies have also optimized the sources of key enzymes. For example, Pang et al. chose two AGEs from *Synechocystis* sp. PCC 6803 (AGE_s_) and *Anabaena* sp. CH1 (AGE_a_), along with three NeuBs from *E. coli* K1, *C. jejuni* NCTC 11168, and the psychrophilic fish pathogen *Moritella viscosa* (NeuB_k_, NeuB_c_ and NeuB_m_). Their findings revealed a 50% increase in Neu5Ac titer in *E. coli* DH5α DN5/pBac carrying AGE_a_ and NeuB_c_ [35]. Zhao et al. selected *neuC* and *neuB* from four microbial sources: *E. coli* O7:K1, *Streptococcus agalactiae*, *C. jejuni*, and *N. meningitidis*. *E. coli* BL21(DE3) carrying *neuC* and *neuB* from *C. jejuni* and *N. meningitidis* produced 4.57 g/L and 4.52 g/L of Neu5Ac, respectively, whereas no Neu5Ac was detected in strains with *neuC* and *neuB* from other sources [22]. In another study on engineering *B. subtilis* for Neu5Ac production, Zhang et al. evaluated key enzymes from various microbial sources and found that AGE from *Anabaena* sp. CH1, and NeuC, NeuB from *N. meningitidis* were more effective than those from other sources [21]. Recently, artificial intelligence (AI) has provided a powerful new paradigm for the efficient mining and screening of enzymes. Ma et al. employed the deep learning model Uni-KP to identify NeuAc synthase with outstanding catalytic activity. This model uses AI techniques and machine learning (ML) algorithms to conduct high-throughput predictions of *k*_cat_ and *K*_m_ values based on protein sequences. According to their study, the newly discovered *Cap*AGE from *Capnocytophaga canimorsus* and *Ir*NeuB from *Idiomarina* sp. Sol25 exhibited significantly higher catalytic efficiency than the previously widely used AGE_a_ and *Nm*NeuB, and further contributed to the production of 70.4 g/L Neu5Ac in *E. coli* [55].

### 2.2. Regulation of Neu5Ac Synthesis in Microbial Systems

In microbial systems, the biosynthesis of Neu5Ac is stringently regulated at both transcriptional and post-transcriptional levels (Figure 3). The expression of *glmS* is under feedback inhibition by GlcN6P, which is accomplished through the mechanisms of small RNAs in *E. coli* and riboswitches in *B. subtilis*. Specifically, in *E. coli*, the *glmS* gene encoding GlmS is post-transcriptionally regulated by two small RNAs, GlmY and GlmZ. When the intracellular concentration of GlcN6P decreases, GlmY and GlmZ accumulate, thereby activating the expression of *glmS* [54,56,57]; in *B. subtilis*, the binding of GlcN6P to *glmS* mRNA activates an intrinsic ribozyme. This ribozyme catalyzes the self-cleavage of *glmS* mRNA, triggering the rapid degradation of *glmS* mRNA by RNase J1 and subsequently shutting down the synthesis of GlmS [58,59,60]. The latest research has shown that in *B. subtilis*, the activity of GlmS is also regulated by GlmR and YvcJ. When the intracellular concentration of UDP-GlcNAc is low, UDP-GlcNAc binds to GlmR and YvcJ, inhibiting the activity of GlmS. Conversely, GlmR directly interacts with GlmS, stimulating its activity [61,62].

Furthermore, the uptake and degradation processes of GlcNAc encoded by the *nag* operon are transcriptionally regulated by NagC, cAMP-CRP, NagR, and GamR. In *E. coli*, the *nagE-nagBACD* operon is under negative regulation mediated by the NagC repressor protein. The binding site of this repressor overlaps with the transcriptional start sites of the *nagE* and *nagB* genes, resulting in the formation of a DNA loop [63,64]. Intriguingly, NagC is also involved in the formation of UDP-GlcNAc and can function as both an activator and a repressor for the transcription of the *glmUS* operon [65]. On the other hand, the expression of the *nag* operon can be activated by the cyclic adenosine monophosphate (cAMP)-cAMP receptor protein (CRP) complex. Moreover, compared to *nagB*, cAMP-CRP has a more pronounced stimulatory effect on *nagE* [65]. In *B. subtilis*, the transport and metabolism of amino sugars encoded by *nagAB*, *nagP*, and *gamAP* [66] are inhibited by the GntR-family repressor proteins NagR and GamR [67]. Specifically, GlcN6P can prevent GamR from binding to all target sites, while NagR can specifically bind to target sites containing the previously identified dre palindromic sequence, and its binding is not inhibited by GlcN6P and GlcNAc6P [67]. Particularly, the Neu5Ac catabolic pathway encoded by the *nanATEK-yhcH* operon naturally exists in *E. coli* [68,69]. This pathway is positively regulated by CRP and negatively regulated by the transcriptional repressor protein NanR encoded by the *nanR* gene located upstream [70,71]. However, owing to the lack of a native Neu5Ac catabolic pathway in *B. subtilis*, this process does not take place in vivo.

In summary, the biosynthesis of Neu5Ac involves a complex regulatory network. Therefore, adopting corresponding metabolic engineering strategies to address the issues in its synthetic pathway, such as enhancing the supply of precursors PEP and ManNAc, reducing the accumulation of by-products like acetic acid and lactic acid in *E. coli* and acetoin in *B. subtilis*, selecting the key enzymes from appropriate microbial sources, and relieving feedback inhibition and repression, is crucial for building an efficient cell factory for Neu5Ac production.

### 2.3. Production Methods and Purification Challenges of Neu5Ac

As market demand keeps rising, cost-effective production of Neu5Ac has drawn growing attention. Comparative evaluation of the main production methods of Neu5Ac is presented in Table 1, including chemical synthesis, enzymatic synthesis, whole-cell catalysis, and microbial fermentation. In recent years, the rapid development of technologies like multi-enzyme cascade, protein engineering, metabolic engineering, and ML has breathed new life into large-scale Neu5Ac production.

Microbial fermentation and whole-cell catalysis, as the most promising industrial production methods for Neu5Ac, still face many challenges in the downstream purification process. How to effectively remove endotoxins from the *E. coli* fermentation broth and substrate residues in the whole-cell catalysis process is crucial for ensuring the safety of biological products and meeting regulatory standards. Chromatography methods, such as ion-exchange chromatography, affinity chromatography, and mixed-mode chromatography, are widely used in the downstream purification of target products from fermentation broths, due to their advantages of high adsorption efficiency and product recovery rates [74]. However, the high cost of chromatographic resins still leads to a relatively high purification cost for Neu5Ac. In the future, efficient and economical purification methods need to be further developed to meet the demand for producing quality products at a decreased cost.

## 3. Strategies in the Construction of Cell Factories for Neu5Ac Production

Owing to the stringent metabolic regulatory mechanisms, the production of Neu5Ac through microbial fermentation has confronted multiple challenges over the past decades, such as feedback inhibition and repression of key enzymes, metabolic flux competition from branch pathways, toxicity from by-product accumulation, precursor shortages, and an imbalance between cell growth and Neu5Ac synthesis. With the development of metabolic engineering, a variety of strategies have been utilized to boost the titer of Neu5Ac, including the application of rational metabolic engineering to relieve feedback regulation, reconfigure metabolic networks, implement dynamic regulation, and optimize carbon sources (Figure 4); as well as the use of irrational strategies including directed evolution of key enzymes and HTS based on biosensors (Figure 5). The combination of rational and irrational metabolic engineering strategies has significantly enhanced the efficient production of Neu5Ac. The main advancements in the production of Neu5Ac using engineered *E. coli* and *B. subtilis* are presented in Table 2.

### 3.1. Rational Metabolic Engineering Strategies

#### 3.1.1. Relieving Feedback Regulation

The synthesis of Neu5Ac in microorganisms is strictly regulated by feedback, including the feedback inhibition of the gene *glmS* by the intermediate product GcN6P and the feedback repression of the *nanATEK* operon by the transcriptional repressor NanR [57,71], which greatly limits the accumulation of Neu5Ac. Therefore, the key to increasing the titer of Neu5Ac lies in relieving feedback inhibition and repression (Figure 4A).

To relieve the feedback inhibition of GlcN6P on *glmS*, many studies have focused on obtaining GlmS variants that are insensitive to this regulation. Within *E. coli*, multi-site mutagenesis of *glmS* (mutant sites: E14K, D386V, S449P and E524G) has been extensively employed in constructing high-titer strains for the production of Neu5Ac [40,76]. The GlmS enzyme is composed of two structurally and functionally distinct domains, the N-terminal domain for glutamine binding and the C-terminal domain for F6P binding [79]. By means of error-prone polymerase chain reaction (ep-PCR), researchers introduced mutations to four amino acids located on the surface of the protein [80]. Interestingly, the mutations of these amino acids substantially decreased the sensitivity of the GlmS enzyme to GlcN6P, enhanced the enzyme’s solubility, yet did not result in the loss of its functionality. X-ray crystallography was utilized to analyze the positions of the *glmS* mutations within the three-dimensional structure of the isomerase domain [81]. The findings indicated that these mutated residues did not belong to the structural components of the GlcN6P binding site and were at a distance of at least 16 Å from GlcN6P. It is likely that the mutations modified the secondary structure of the polypeptide backbone or the local side-chain interactions, thereby exerting an indirect influence on the binding of GlcN6P [80]. The expression of the *glmS* quadruple-site mutant in *E. coli* DH5α led to a 37% increase in the titer of Neu5Ac compared to the wild-type strain [76].

In the process of constructing Neu5Ac engineering strains, it is common to delete the *nanATEK* operon to relieve the feedback repression of NanR on related genes, which has successfully increased the titer of Neu5Ac [40]. Additionally, exploring new pathways to replace feedback repression reactions is also a common strategy. For instance, in *E. coli*, by heterologously introducing the *age* and *neuB* genes to construct the AGE pathway, or by heterologously introducing the *neuC* and *neuB* genes to construct the NeuC pathway, researchers have achieved Neu5Ac titers of 16.7 g/L [35] and 58.26 g/L [22], respectively. In brief, through introducing enzyme mutants insensitive to feedback inhibition, knocking out genes associated with feedback repression, and building new alternative synthetic pathways, feedback inhibition and repression have been relieved, which has become a common strategy for the construction of Neu5Ac cell factories.

#### 3.1.2. Decreasing the Uptake and Degradation of Neu5Ac

In *E. coli*, the four crucial proteins encoded by the *nanATEK* operon exert a direct impact on the degradation of Neu5Ac [68], thereby resulting in a decrease in its accumulation level. During this metabolic process, Neu5Ac is translocated across the cell membrane into the intracellular space by the sia transporter NanT and Neu5Ac outer membrane channel protein NanC [82,83]. Subsequently, it is reversibly hydrolyzed by NanA into ManNAc and pyruvate [84]. Thereafter, through the catalytic actions of NanK and NanE, ManNAc is degraded to form GlcNAc6P [85]. The combinatorial deletion of the *nanATEK* genes, aimed at attenuating the degradation of Neu5Ac, has been extensively utilized in the development of high-performance engineered strains for Neu5Ac production (Figure 4B). For example, Zhao et al. used pETDuet-1 as a vector to introduce *C. jejuni neuC* and *neuB* into *E. coli* BL21(DE3), constructing the Neu5Ac synthetic pathway. However, the generated *E. coli* BL21(DE3) EM1 did not produce detectable Neu5Ac. Apparently, the existence of the degradation pathway hindered the biosynthesis of Neu5Ac. After further deletion of the *nanATK* operon, the synthesis of Neu5Ac was successfully achieved [22]. Similarly, in the case of *E. coli* DH5α DN5/pNnsGM, where *nanATEK* was simultaneously deleted, the titer of Neu5Ac represented a 19% increase compared to *E. coli* DH5α DN4/pNnsGM [76]. In conclusion, blocking the uptake and degradation pathways is a fundamental step in constructing high-titer Neu5Ac strains, which is of great significance for the initial realization of in vivo production of Neu5Ac in *E. coli*.

#### 3.1.3. Reducing the Accumulation of By-Products

Acetic acid and lactic acid are two prevalent acidic by-products during the *E. coli* fermentation process. Their overproduction can impede the growth of recombinant strains and result in a decline in the synthesis efficiency of Neu5Ac. Employing gene knockout to eliminate competing pathways and redirect metabolic fluxes towards product synthesis represents an effective approach to tackle these problems (Figure 4C). In *E. coli*, typically, two major pathways for acetic acid production are concurrently blocked: the phosphotransacetylase-acetate kinase (*pta*-*ackA*) pathway and the pyruvate oxidase (*poxB*) pathway [86]. For example, in *E. coli* DH5α DN1/pNnsGM, after the deletion of *ackA*, the secretion of acetic acid was reduced by 67%. Nevertheless, upon further deletion of *poxB*, a decrease in the Neu5Ac titer was observed. This is because the prior inactivation of the Pta-AckA pathway led to a substantial increase in lactic acid secretion [87]. Consequently, by further deleting the gene *ldhA* encoding lactate dehydrogenase, *E. coli* DH5α DN4/pNnsGM was obtained, in which the Neu5Ac titer increased from 0.3 g/L to 1.36 g/L and no lactic acid was produced [76]. Distinct from *E. coli*, *B. subtilis* synthesizes acetic acid solely via the Pta-AckA pathway. Moreover, due to the competition from the acetoin synthesis pathway, the accumulation of acetic acid in *B. subtilis* is far lower than that in *E. coli*. Deletion of *pta* and *ldhA* can effectively mitigate the accumulation of organic acids [77]. It is noteworthy that although the overflow of the neutral by-product acetoin, encoded by the *alsR*-*alsSD* operon, does not cause acidification of the cellular growth environment, it competes with the Neu5Ac synthesis pathway for carbon flux [88]. However, up to the present, no reports have been published regarding strategies for modifying this operon in *B. subtilis* to construct Neu5Ac-engineering strains. This may be because neutral acetoin helps maintain pH during the fermentation process, preventing cytoplasmic acidification that would damage strain growth [52]. This may well be a potential research focus.

Although the deletion of the Pta-AckA pathway can substantially reduce the generation of acetic acid, this alteration may give rise to intracellular redox imbalance and energy dissipation [89,90]. Such disruptions to cellular homeostasis can still lead to a decline in the growth rate of the strain, thereby exerting an adverse impact on the efficient production of Neu5Ac. To overcome these limitations, strategies leveraging cofactor engineering to balance redox reactions and mitigate acetic acid accumulation have been investigated. Based on alleviating the transcriptional repression of aerobic functional genes by the global regulatory factor ArcA, the *iclR* gene was further knocked out to activate the glyoxylate cycle. Additionally, the *nox* gene (derived from *Streptococcus mitis*) encoding NADH oxidase was heterologously introduced and overexpressed. In comparison with the parental *E. coli* BL21(DE3) NBC45, the resultant *E. coli* BL21(DE3) BG02 exhibited a 3.37-fold reduction in acetic acid accumulation, accompanied by an almost 3-fold increase in Neu5Ac production [38]. Moreover, maintaining the supply of ATP is of equal significance for enhancing cell growth and Neu5Ac production. Heterologously introducing *Seppk* (derived from *Staphylococcus epidermidis*) as an ATP-supplying module and overexpressing it in *E. coli* led to a 27% increase in OD_600_ and a 12% increase in the titer of Neu5Ac [38]. In a recent study, Sun et al. deleted the *rhaB* gene encoding L-rhamnose kinase to eliminate an ATP-consuming futile cycle, resulting in a 77.6% increase in ATP concentration and a 31% reduction in acetic acid in *E. coli* W3110 NEA-27 [27]. Notably, after 56 h of fermentation at 30 °C in a 5 L bioreactor, *E. coli* W3110 NEA-27 achieved a Neu5Ac titer of 77.12 g/L [27]. This represents the highest reported titer for the production of Neu5Ac through microbial fermentation and serves as a paradigmatic example of applying cofactor engineering to boost functional sugar titers. This further underscores the significance of tightly regulating redox balance and energy-supply modules in minimizing by-products accumulation and generating highly efficient and robust Neu5Ac-producing engineered strains. However, the current research in this domain remains limited. Thus, it is imperative to explore a broader range of innovative cofactor-balancing (ATP, NADH) strategies. For example, coordinated reprogramming of ATP metabolism [91], the augmentation of the biosynthesis of endogenous cofactor pools [92], and the establishment of novel artificial cofactor systems [93].

#### 3.1.4. Enhancing the Supply of Precursors

##### Optimizing Synthetic Pathways to Enhance ManNAc Supply

ManNAc functions as one of the pivotal precursors in the synthesis of Neu5Ac [94] (Figure 4D). On one hand, obstructing the degradation pathway of ManNAc is of paramount importance for the accumulation of Neu5Ac. Both GlcNAc and UDP-GlcNAc can be catalytically converted into ManNAc within the cell. Reducing the catabolism of GlcNAc by combined deletions of the *nanEBAC* and *nagBACD* operons is a common approach to increase the supply of GlcNAc to ManNAc. For instance, *E. coli* DH5α/pNnsGM hardly accumulates GlcNAc and ManNAc. After further deletion of the genes *nagA* and *nagB*, however, the generated *E. coli* DH5α DN1/pNnsGM shows a significant increase in the accumulation of GlcNAc and ManNAc, reaching 6.48 g/L and 1.55 g/L, respectively [76]. Moreover, the enzyme encoded by *wecB* is capable of isomerizing UDP-GlcNAc into UDP-*N*-acetyl-mannosamine (UDP-ManNAc), significantly hampering the accumulation of UDP-GlcNAc. Deletion of the *wecB* gene is also a commonly adopted strategy to reduce the catabolism of ManNAc [40].

On the other hand, enhancing the biosynthesis of ManNAc is of great significance for boosting the metabolic flux of Neu5Ac. Generally, Neu5Ac engineered strains produce ManNAc via two pathways: the GlcNAc6P pathway and the GlcNAc1P pathway. In the GlcNAc6P pathway, many investigations employing *E. coli* as the chassis cell have utilized the reversible enzyme encoded by *slr1975* (derived from *Synechocystis* sp. PCC6803) to catalyze the conversion of GlcNAc into ManNAc. However, this enzyme exhibits a relatively low affinity for ATP and is inhibited by an excessive amount of pyruvate, thereby restricting its catalytic efficiency [95]. Compared with *slr1975*, *age* (derived from *Anabaena* sp. CH1), which features sufficient substrate affinity and higher catalytic activity, was a promising alternative. To enhance this conversion, Pang et al. optimized the expression intensity of AGE and NeuB through RBS engineering, leading to a 1.04-fold increase in Neu5Ac production [35]. Furthermore, based on *B. subtilis* No. 6, the expression level of *GNA1* was upregulated by using an engineered NCS, which increased the Neu5Ac titer of *B. subtilis* BgG-N *abrB*-30bp by 129% [36].

Most of the latest research has been centered on improving the supply module of ManNAc in the GlcNAc1P pathway. Through deletion of *nagAB*, *nanATEK*, and *manXYZ* to block ManNAc catabolism, and introduction of the GlmS quadruple-site mutant (GlmS*) to relieve GlcN6P feedback inhibition, the ManNAc titer in *E. coli* BL21(DE3) NBC2 rose by 0.40 g/L compared to the wild-type strain [40]. In this pathway, there are two approaches to strengthening the supply of ManNAc: optimizing the expression level of the exogenous NeuC enzyme and increasing the supply of the precursor UDP-GlcNAc. Through the construction of NeuC expression cassettes with different strengths, it was discovered that ManNAc was only produced when a low-intensity promoter was expressed in *E. coli* BL21(DE3) NBC3. When the weakest promoter P1 was applied, the ManNAc titer reached its highest value of 1.25 g/L. Furthermore, as a key precursor of ManNAc, UDP-GlcNAc can be synthesized in *E. coli* from F6P via consecutive catalytic reactions of three endogenous genes, *glmS*, *glmM*, and *glmU*. Previous studies have indicated that co-overexpressing *glmMUS* using a high-copy-number vector might be more conducive to enhancing the supply of the precursor UDP-GlcNAc than using medium/low-copy-number vectors [96]. Consequently, by overexpressing *glmMUS* using the pRSFDuet-1 vector, the Neu5Ac titer of *E. coli* BL21(DE3) EM9 reached 4.53 g/L, which was significantly higher compared to when the pETDuet-1 vector was used [22], thereby verifying the above-mentioned view. However, plasmid-based strategies tend to impose an additional metabolic burden. At the genomic level, the supply of the UDP-GlcNAc precursor has been partitioned into two modules: the GlmM and GlmU-GlmS* expression modules, followed by global optimization. It was found that the *E. coli* BL21(DE3) NBC16, which combined a low expression level of GlmM with a medium expression level of the GlmU-GlmS* module, exhibited the highest ManNAc titer of 8.95 g/L [40]. Interestingly, overexpressing GlmM or GlmU alone had contrasting effects on Neu5Ac synthesis. The former slightly promoted it, while the latter significantly decreased the Neu5Ac titer [21], which serves as a reminder that a rational design of the combination of enzyme expression levels is essential. Overall, strengthening the supply of ManNAc, such as blocking its degradation pathway and increasing its biosynthesis level, is of vital importance for enhancing Neu5Ac production.

##### Rewiring the Central Metabolic Module to Increase PEP Availability

PEP functions as another crucial precursor in the biosynthesis of Neu5Ac (Figure 4D). In several research studies, the irreversible enzyme NeuB has been frequently employed to replace NanA. This reaction, utilizing ManNAc as the direct precursor and PEP as the co-substrate, facilitates the conversion to Neu5Ac. The reason is that in the natural biosynthetic pathway, the NanA-catalyzed reaction converting ManNAc and pyruvate into Neu5Ac is reversible. Importantly, the reaction equilibrium favors the cleavage of Neu5Ac, ultimately reducing the titer of Neu5Ac [84]. When *E. coli* grows in a minimal medium with glucose as the carbon source, the primary metabolic pathways of PEP are as follows: PTS consumes about 50% of the total PEP, and the reactions catalyzed by PEP carboxylase, pyruvate kinases, UDP-*N*-acetylglucosamine enolpyruvyl transferase, and 3-Deoxy-d-arabino-heptulosonate 7-phosphate (DAHP) synthase consume approximately 16%, 15%, 16% and 3%, respectively [97]. Deletion of select genes encoding crucial nodes in PEP consumption can effectively augment the supply of PEP, thereby enhancing the production of Neu5Ac.

In a significant number of Gram-negative and Gram-positive bacteria, the PTS functions as the primary sugar transport mechanism [98]. Therefore, blocking the PTS can potentially be an effective strategy for conserving PEP. This argument was further corroborated by the genetic deletion of *ptsG* in *E. coli* DH5α DN11, which led to a marked increase in Neu5Ac titer from 9.9 g/L to 16.7 g/L [35]. In *B. subtilis*, glucose can be directly absorbed via the transporter proteins GlcP and GlcU. Metabolic engineering strategies that design non-PTS pathways (*glcp*, *glck*) to enhance glucose uptake efficiency also have successfully elevated Neu5Ac titers [21,77]. Another crucial strategy for promoting PEP accumulation involves obstructing the glycolytic pathways at the F6P and PEP nodes [99]. By further deleting *pykA* from *E. coli* DH5α DN5, the resulting strain DN11 showed an increase in the Neu5Ac titer from 3.7 g/L to 9.9 g/L [35]. Furthermore, modifying the oxaloacetate replenishment pathway and alleviating the process by which citrate synthase (*gltA*) drives acetyl-CoA into the TCA cycle may also contribute to increased PEP accumulation by preventing its degradation [35].

PEP serves as a critical metabolic node requiring precise regulation. Although the aforementioned strategies have all achieved certain effects, forcibly blocking PEP-consuming pathways may disrupt metabolic balance and impede both strain growth and Neu5Ac synthesis. For example, based on *E. coli* DH5α DN5, single deletion of *ppc*, *pykF* or *gltA* all severely impaired growth, likely due to essential metabolite deficiency from pathway disruptions; double deletion of *gltA* and *pykA* significantly reduced the Neu5Ac titer, as it substantially restricted the TCA cycle, causing insufficient energy supply [35]. Consequently, to achieve more precise modulation of the metabolic flux at the PEP node, a recent study employed CRISPR interference (CRISPRi) technology in conjunction with the orthogonal inducible promoters P_lacO1_, P_LtetO-1_, and P_araBAD_ in *E. coli*. Through optimal combinatorial inhibition of *ptsI* (encoding PTS-related proteins), *pta*, and *pykA*, the titer of Neu5Ac was enhanced by 2.4-fold [100]. In summary, enhancing the supply of the PEP precursor pool by optimizing central metabolic modules, such as the PTS, glycolytic pathway, and oxaloacetate replenishment pathways, constitutes an effective strategy for boosting the production of Neu5Ac.

#### 3.1.5. Dynamic Regulation

Among the aforementioned metabolic engineering approaches, a substantial proportion employ static regulatory strategies, primarily involving gene knockout or constitutive overexpression. Nevertheless, this inevitably leads to an imbalance in cell growth and Neu5Ac production, thereby limiting the cell factory’s production capacity. To address this issue, efforts have focused on constructing regulatory elements and designing genetic circuits to enable real-time adjustment of metabolic fluxes based on the cell growth stage, facilitating the development of Neu5Ac-engineered strains.

In *B. subtilis*, several dynamic control systems have been designed to precisely regulate metabolic fluxes or enhance the robustness of production (Figure 4E). NCSs serve as crucial tools for the fine-tuning of gene expression levels. They regulate gene expression at the translational level by influencing the binding efficiency between ribosomes and mRNA, as well as the elongation of ribosomes during the translation initiation stage [101]. Based on kinetic modeling, researchers have exploited growth-coupled NCSs to dynamically downregulate the expression of *pyk*. This allows normal translation during the exponential growth phase and a decline during the stationary phase. When compared to simply disrupting the conversion of PEP to pyruvate by deleting *pyk*, this dynamic regulatory strategy not only ensures an adequate supply of PEP but also significantly alleviates the negative impact on strain growth. Consequently, the titer of Neu5Ac has been increased by 3.21-fold [36].

Furthermore, dynamic regulation has been successfully implemented to modulate the producing and nonproducing subpopulations of engineered *B. subtilis* during fermentation. In this context, researchers have designed an inducible population quality control system (iPopQC) to enhance the robustness and efficiency of biological production. Specifically, iPopQC utilizes a Neu5Ac-responsive biosensor to regulate the expression levels of the growth-essential genes *folB* and *ftsW* under the control of the inducible promoter P_veg105_. By coupling cell growth with Neu5Ac synthesis, iPopQC confers a growth advantage to the producing cell subpopulations. It is noteworthy that this coupling, which persists in the fermentation process when the traditional population quality control system (PopQC) is employed [102], may be detrimental to early-stage product synthesis. In contrast, iPopQC can dynamically adjust the cell subpopulations by varying the timing of inducer (IPTG) addition, thereby reducing such adverse effects. In fact, through enriching the producing cell subpopulation during fermentation, iPopQC has increased the Neu5Ac titer by 1.97-fold compared to the control group without IPTG addition, and by 52% compared to the static cell subpopulation group (PopQC group) [37].

In *E. coli*, unlike the scenario in *B. subtilis*, within the literature we surveyed regarding Neu5Ac production, few reports have been found on the application of dynamic regulation strategies as far as we know. To optimize Neu5Ac production, further exploration of this strategy is warranted. Overall, dynamic regulation offers an efficient and robust approach for enhancing Neu5Ac biosynthesis, effectively minimizing the growth defects and production losses associated with genetic engineering.

#### 3.1.6. Carbon Source Optimization

##### Single Carbon Source

Carbon sources play a pivotal role in the growth and metabolism of microbial cells. Their utilization exerts a profound influence on the synthesis efficiency of target products within microbial cell factories. Currently, the maximum titers of Neu5Ac synthesized with *E. coli* or *B. subtilis* as chassis cells have reached 77.12 g/L [27] and 30.10 g/L [21], respectively, and both utilized glucose as the carbon source. In research regarding the production of Neu5Ac with a single carbon source, glucose has been used more frequently than glycerol. However, engineered strains constructed based on different metabolic strategies possess distinct characteristics. Thus, the selection of a suitable carbon source is of great significance. For example, by comparing the impacts of three groups of carbon sources (100% glycerol, 100% glucose, and a mixture of 50% glycerol and 50% glucose) on the Neu5Ac synthesis efficiency of *E. coli* BL21(DE3) EM19, it was found that when glycerol served as the sole carbon source, the Neu5Ac titer was the highest, reaching 46.92 g/L [22]. This may be attributed to the unique metabolic properties of glycerol. For instance, compared to using glucose as a carbon source, the formation of acetic acid in *E. coli* is more limited when glycerol is used [103]. Another possible reason is that the conversion of glycerol to PEP requires fewer enzymes than the conversion of glucose to PEP [48]. This implies that glycerol metabolism imposes a lower metabolic burden compared to glucose metabolism, which is conducive to PEP synthesis. Furthermore, different carbon sources may lead to varying supplies of ManNAc. It has been reported that using glycerol as a carbon source can result in nearly twice as much ManNAc accumulation as using glucose [40].

##### Mixed Carbon Sources

In certain scenarios, compared with the utilization of a single carbon source, the employment of mixed carbon sources proves to be highly advantageous for industrial microbial production [104]. Through the combined provision of carbon sources, precursors of biomass components can be more effectively enriched, thereby facilitating cell growth [105]. Moreover, by co-utilizing carbon sources, it becomes possible to re-balance cell growth and the biosynthesis of target products, thus attaining higher productivity [106]. At present, mixed carbon sources have been incorporated into the construction of engineered *E. coli* and *B. subtilis* for Neu5Ac production. For example, glucose and malate are used to balance the supply of key precursors GlcNAc and PEP [77], and glucose and glycerol are used to increase the accumulation of PEP [78].

The binary carbon source composed of glucose and malic acid can be inherently and distinctively co-utilized by *B. subtilis* [107] (Figure 4F). This renders it feasible to modulate the precursor supply via modular pathway engineering, thereby achieving an increase in Neu5Ac production. Specifically, glucose is employed for the GlcNAc supply module, while malic acid is dedicated to the PEP supply module. To strike a balance between product synthesis and cell growth, several modifications were made to the utilization pathways of glucose and malic acid. Glycolysis was obstructed by deleting the *pyk* gene. The Entner–Doudoroff (ED) pathway was introduced through the overexpression of *edd* and *eda*, and *ytsJ* was overexpressed. Simultaneously, *glmS* and *yqaB* were overexpressed to reinforce the GlcNAc supply. Additionally, *ptsG* was deleted and *pckA* was overexpressed to augment the PEP supply. As a result, the titer of Neu5Ac increased by 13.6-fold, reaching 2.18 g/L [77].

The co-utilization of glucose and glycerol has also been exploited in the construction of engineered *E. coli* and *B. subtilis* for Neu5Ac production. This approach aims to elevate the intracellular PEP concentration, thereby optimizing the synthesis efficiency of Neu5Ac. In *E. coli*, Ma et al. relieved carbon catabolite repression by disrupting the PTS and introduced *glpK** to enhance glycerol utilization, establishing a synergetic carbon utilization system. When the ratio of glucose to glycerol was 1:2, the Neu5Ac titer of *E. coli* BL21 (DE3) BLNK-5 reached 70.4 g/L after 60 h of fermentation in a 5 L bioreactor [55]. In *B. subtilis*, the catabolic pathway of glycerol predominantly encompasses three enzymes: glycerol transport facilitator (GlpF), glycerol kinase (GlpK), and glycerol-3-phosphate dehydrogenase (GlpD). Through the catalytic action of GlpD, glycerol can gain entry into the Embden–Meyerhof–Parnas pathway, thereby facilitating the supply of PEP [108]. However, due to the glucose catabolite repression mechanism in *B. subtilis*, glycerol cannot serve as a carbon source in the presence of glucose. To overcome this challenge, *glpK* was overexpressed under the regulation of the P_odhA_ promoter. Subsequently, with glucose and glycerol serving as dual carbon sources, *B. subtilis* B8 was subjected to shake-flask fermentation. As a result, the PEP concentration increased by 33.6%, and the titer of Neu5Ac was further enhanced [78]. Briefly, optimizing the carbon source to increase the supply of precursors is a promising strategy for enhancing the titer of Neu5Ac.

### 3.2. Irrational Strategies

Directed enzyme evolution (Figure 5A) and HTS based on Neu5Ac biosensors (Figure 5B) are common irrational strategies for constructing efficient Neu5Ac cell factories. In prior research, directed evolution, which involves generating mutant libraries via random mutagenesis followed by screening, has been extensively employed to enhance the catalytic activity of key enzymes and optimize the Neu5Ac synthesis pathway, thereby improving the performance of microbial cell factories [38,39]. However, the efficient identification of desired phenotypes from highly diverse mutant libraries has emerged as a significant bottleneck in the practical implementation of this strategy. Consequently, the development of novel HTS system for the rapid and effective identification of Neu5Ac-producing strains with superior performance is of utmost necessity.

In recent years, biosensors have demonstrated substantial potential in HTS applications [109]. Researchers combined a biosensor based on Neu5Ac-specific aptamer enzymes with a Ni^2+^-based growth-coupled screening system to establish a HTS platform, which enables cells accumulating more Neu5Ac to survive in the presence of Ni^2+^. This platform effectively optimized the Neu5Ac biosynthesis pathway and conducted directed evolution on the pivotal enzyme NeuB, which improved the titer of Neu5Ac by 34% and 23%, respectively [32]. However, this biosensor only works soundly at 0–86.4 ± 5.1 mg/g DCW of intracellular Neu5Ac [32]. To further enhance the performance of this biosensor, in vivo evolution and screening of riboswitches were performed. This was also achieved through growth-coupled screening, incorporating both positive (Ni^2+^) and negative (tetracycline) selection mechanisms. Consequently, the threshold and dynamic range of the riboswitch-based biosensor were significantly enhanced, reaching 0–18.38 g/L Neu5Ac and 3.12-fold, respectively. Subsequently, the newly developed riboswitch was utilized to evolve the Neu5Ac synthesis pathway and the key enzyme AGE. The desired mutant, *E. coli* DH5α DN5/pBac2B, was isolated via fluorescence screening. The Neu5Ac production of this mutant reached 14.23 g/L, representing a 42% increase compared to the use of the original riboswitch [39].

In addition, transcription-factor-based biosensors have also been extensively investigated and applied in the screening of high-titer Neu5Ac strains. The Neu5Ac-responsive biosensor was usually developed based on the transcriptional regulator NanR. NanR specifically interacts with Neu5Ac and binds to its regulatory region, thereby alleviating the transcriptional repression of downstream genes [110]. In a recent study, the NanR regulator from *Bifidobacterium breve* was employed to elevate the expression of the repressor protein. Subsequently, the specific recognition sequence RiboJ was introduced, further enhancing the biosensor’s sensitivity. This extended the biosensor’s detection range for extracellular Neu5Ac to 0–20 g/L, with a maximum activation ratio of 10.62-fold. By integrating this highly sensitive Neu5Ac-responsive biosensor with flow cytometry, a random mutant library was constructed by modulating the combination of ribosome-binding sites (RBS), spacer sequences, and start codons. HTS was then carried out, accompanied by the concurrent optimization of the expression levels of *glmS**-*glmM*-*SeglmU*. Eventually, the highly efficient Neu5Ac-producing strain *E. coli* BL21(DE3) A39 was successfully obtained, achieving a Neu5Ac titer of 58.26 g/L in a 3 L bioreactor [38]. It is worth noting that the strains initially screened by biosensors carry the risk of false positives or negatives. Generally, they will be further verified through several rounds of well plate or shake flask fermentation, so as to obtain the strain with the optimal Neu5Ac titer. In conclusion, the integration of biosensors with HTS technologies holds great promise for optimizing the Neu5Ac biosynthesis pathway, guiding the directed evolution of key enzymes, and efficiently screening mutants with enhanced production capabilities. This approach is expected to emerge as one of the most effective means of developing high-titer Neu5Ac strains that meet industrial requirements.

## 4. Conclusions and Perspectives

Given the escalating commercial utilization of Neu5Ac across the food, pharmaceutical, and cosmetic areas, there has been a sustained and significant surge in market demand for its industrial-scale production. In recent years, propelled by the concept of sustainable development and green-oriented policies, the production of Neu5Ac via microbial fermentation has drawn substantial attention. Remarkable advancements have been achieved in the construction of efficient cell factories for Neu5Ac synthesis. This review comprehensively examines the synthetic pathways and regulatory mechanisms underlying Neu5Ac production in *E. coli* and *B. subtilis*, along with its upstream production methods and downstream purification challenges. Particular emphasis was placed on the rational and irrational strategies in the cell engineering construction of Neu5Ac. Rational strategies include: (1) alleviating feedback regulation; (2) reducing the uptake and degradation of Neu5Ac; (3) decreasing the accumulation of by-products; (4) enhancing the supply of precursors; (5) implementing dynamic regulation; (6) altering the carbon source to optimize fermentation conditions. Irrational strategies include: (1) directed enzyme evolution; (2) HTS based on Neu5Ac biosensors. These strategies offer valuable insights and serve as a reference framework for future Neu5Ac production endeavors.

Despite the fact that the manipulation of *E. coli* or *B. subtilis* through metabolic engineering has, to a certain degree, augmented the production of Neu5Ac, substantial challenges persist in its large-scale industrial production. For instance, the decoupling of cell growth from Neu5Ac synthesis has yet to be accomplished; the intricate regulatory networks within engineered strains still demand further elucidation; and the fermentation process remains in need of more in-depth optimization. In recent times, the progress of synthetic biology and systems metabolic engineering [111] has presented viable solutions to these challenges. Looking ahead, the following strategies can be implemented to develop high-performance engineered strains for Neu5Ac production: (1) Leveraging extensive omics data, including genomics, transcriptomics, proteomics, and metabolomics, in conjunction with computational models, machine learning, and artificial intelligence techniques, to rationally design metabolic pathways and accurately predict their performance. This approach aims to alleviate metabolic burden and enhance the efficiency of product synthesis [112,113]; (2) Adopting dynamic regulation strategies, whether metabolite-specific (involving transcription factors and biosensors) or non-specific (notably environmental regulation, growth-phase responses, and quorum sensing), to further balance the precursor supply between cell growth and product synthesis [114,115]; (3) Prior to scaling up production from laboratory to industrial levels, optimizing fermentation process variables such as macronutrients, micronutrients, inducers, growth factors, and temperature. This optimization is crucial for maximizing growth and productivity [116,117]. In the future, to better promote the development of the Neu5Ac industry, it is essential to further explore how to achieve both industrial scalability and economic feasibility. This involves a multifaceted approach, including the optimization of metabolic pathways and fermentation processes, and the innovation of downstream purification technologies. In conclusion, using the advanced strategies to construct high-performance strains can increase Neu5Ac titer, reduce costs, and meet industrial needs.

## Figures and Tables

**Figure 1 foods-14-03478-f001:**
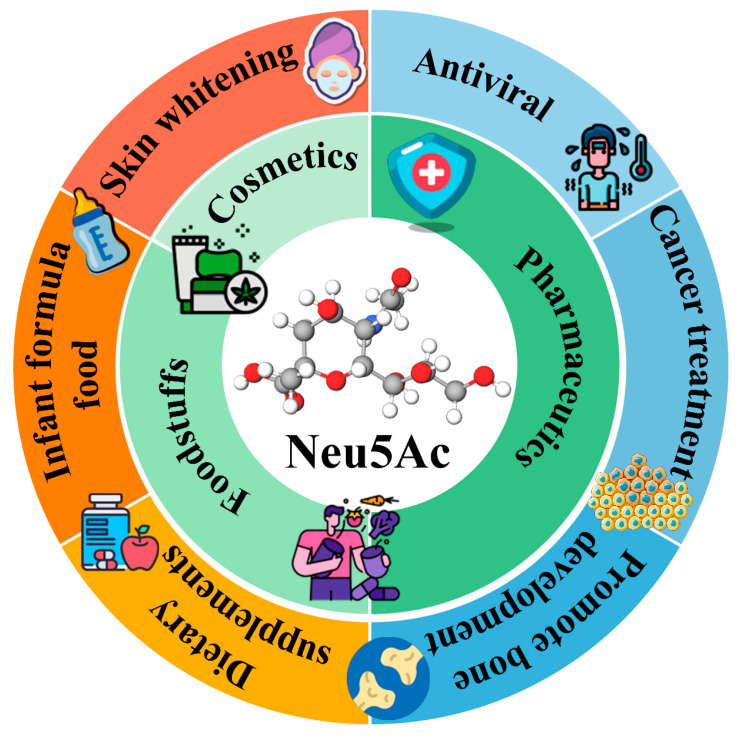
Physiological functions and applications of Neu5Ac. Neu5Ac, with its multifunctional biological characteristics, is widely applied in the fields of functional foods, medicine and cosmetics.

**Figure 2 foods-14-03478-f002:**
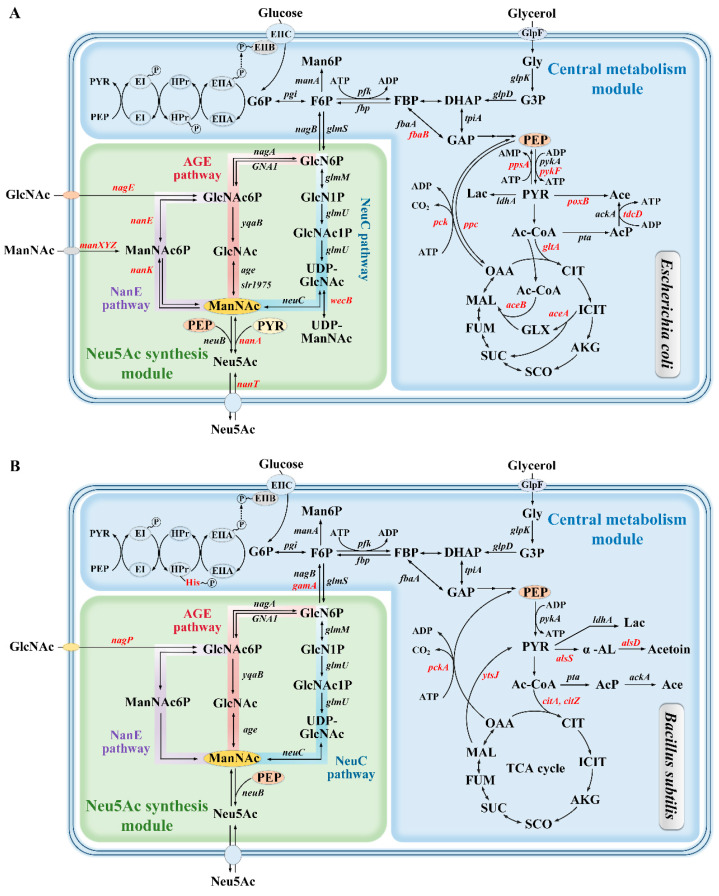
Metabolic pathways of Neu5Ac in *E. coli* and *B. subtilis*. (**A**) Biosynthesis of Neu5Ac in *E. coli*. (**B**) Biosynthesis of Neu5Ac in *B. subtilis*. The blue background represents the central metabolic module, and the green background represents the Neu5Ac synthesis module, which includes the NanE pathway (purple lines), the AGE pathway (red lines), and the NeuC pathway (blue lines). In metabolic pathways, the differences between *E. coli* and *B. subtilis* are marked in red. On the cell membrane, the ovals represent transport proteins. Major metabolite abbreviations are as follows: Neu5Ac: *N*-Acetylneuraminic acid; Gly: glycerol; G3P: glycerol-3-phosphate; DHAP: dihydroxyacetone phosphate; GAP: glyceraldehyde 3-phosphate; FBP: fructose 1,6-bisphosphate; F6P: fructose 6-phosphate; G6P: glucose-6-phosphate; PEP: phosphoenolpyruvate; PYR: pyruvate; Lac: lactate; α-AL: acetolactate; Man6P: mannose 6-phosphate; GlcN6P: glucosamine 6-phosphate; GlcN1P: glucosamine 1-phosphate; GlcNAc1P: *N*-acetylglucosamine 1-phosphate; UDP-GlcNAc: uridine 5′-diphospho-*N*-acetylglucosamine; UDP-ManNAc: uridine 5′-diphospho-*N*-acetylmannosamine; GlcNAc6P: *N*-Acetylglucosamine-6-phosphate; GlcNAc: *N*-acetylglucosamine; ManNAc: *N*-acetylmannosamine; ManNAc6P: *N*-acetylmannosamine 6-phosphate.

**Figure 3 foods-14-03478-f003:**
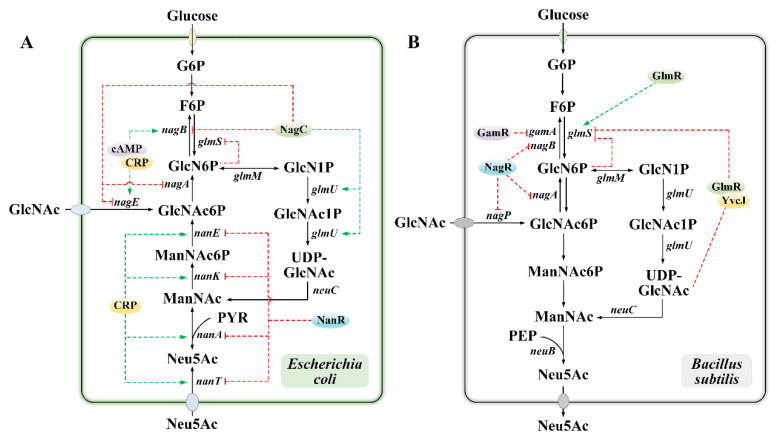
Regulatory mechanisms of the Neu5Ac biosynthesis pathway in *E. coli* and *B. subtilis*. (**A**) Regulation of Neu5Ac biosynthesis in *E. coli*. (**B**) Regulation of Neu5Ac biosynthesis in *B. subtilis*. In the figure, feedback inhibition and repression are indicated by red dashed lines, activation by green dashed lines.

**Figure 4 foods-14-03478-f004:**
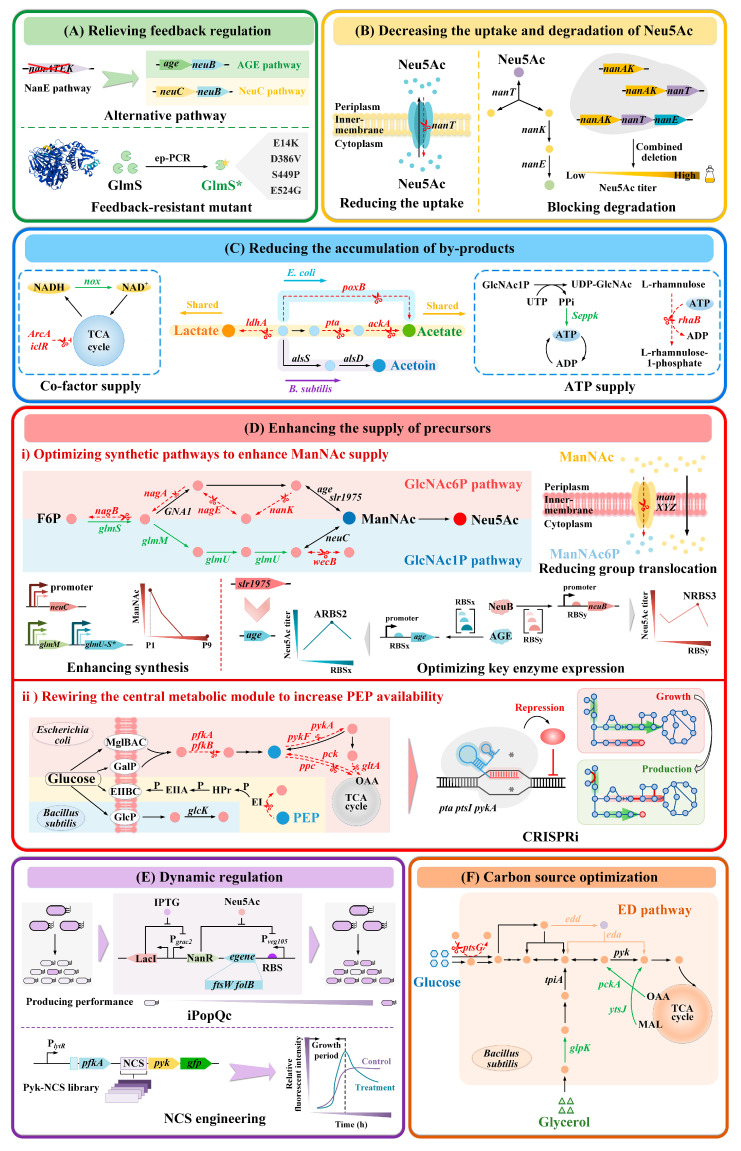
The application of rational metabolic engineering strategies in Neu5Ac production. (**A**) By building alternative pathways and constructing GlmS variants to alleviate feedback regulation; (**B**) Decreasing the uptake and catabolism of Neu5Ac; (**C**) Blocking by-product pathways, maintaining NADH/NAD^+^ balance, and enhancing ATP supply contribute to the reduction in acetic acid and lactic acid accumulation; (**D**) Enhance the supply of precursors, including: (i) optimizing synthetic pathways to enhance ManNAc supply; (ii) rewiring the central metabolic module to increase PEP availability; (**E**) The inducible population quality control system and N-terminal coding sequence engineering are used to dynamically regulate cell production/nonproduction subpopulations and metabolic fluxes, respectively, achieving a balance between cell growth and Neu5Ac production; (**F**) Further increasing the titer of Neu5Ac by optimizing the carbon sources. In the figure, gene knockout is indicated by red dotted lines, gene overexpression by green lines, and mutants by “*”.

**Figure 5 foods-14-03478-f005:**
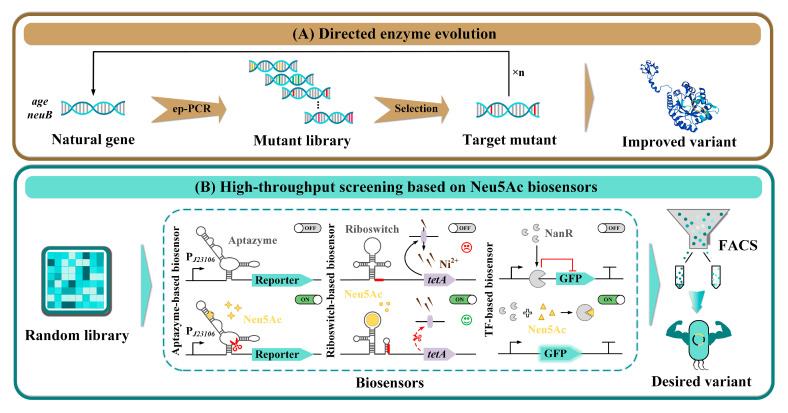
The application of irrational strategies in the construction of cell factories for Neu5Ac production. (**A**) Optimizing key enzymes such as NeuB and AGE through directed evolution; (**B**) Use biosensors, such as aptazyme-based, riboswitch-based, and transcription factor-based systems, to conduct high-throughput screening of the strain mutant library.

**Table 1 foods-14-03478-t001:** Comparative evaluation of the main production methods of Neu5Ac.

Method	Cost	Yield	Scalability	Safety	Representative Reference
Chemical synthesis	High costs mainly result from complex reactions, pricey catalysts, and strict purification.	Involves multi-step reactions and stereochemical control, resulting in low yield.	Cumbersome protection steps and reaction conditions make it unsuitable for large-scale production.	High safety risks; eco-unfriendly.	[16]
Enzymatic synthesis	The preparation and purification of enzymes, along with the requirement for ATP addition, result in high costs.	High conversion rate, with the overall conversion rate of the two-step enzymatic catalytic reaction reaching up to 82%.	Immobilized enzyme technology offers the possibility of large-scale production, but enzyme stability and cost are major limiting factors.	Low biosafety risks; eco-friendly.	[72]
Whole-cell catalysis	Saves enzyme purification costs, but requires adding pricey and excess substrates (like pyruvate, GlcNAc), at medium cost.	High yield, GlcNAc conversion maxes at 77%, via *Bacillus amyloliquefaciens* dual-cell co-catalysis.	Shows good scalability potential, but costs are relatively high for large-scale industrial use.	Biosafety relies on host strain used; high operational safety.	[73]
Microbial fermentation (*E. coli*)	Often use cheap carbon sources (like glucose, glycerol) as raw materials, with low costs.	High-yield Neu5Ac production at 0.217 g/g glucose has been reported.	High scalability, since mature fermentation technology fits large-scale industrial output.	Biosafety risks exist, as *E. coli* contains endotoxins.	[27]
Microbial fermentation (*B. subtilis*)		Relatively high-yield Neu5Ac production at 0.049 g/g glucose has been reported.		Highest safety due to endotoxin-free *B. subtilis* (GRAS).	[21]

**Table 2 foods-14-03478-t002:** Advancements in fermentative Neu5Ac production via engineered *Escherichia coli* and *Bacillus subtilis*.

Strain	Construction Strategies	Cultivation	Carbon Sources	Titer (g/L)	Reference
*E. coli* MG1655	Δ*nanT*, Δ*nanA*, (+)*glmS*, (+)*neuB*, (+)*neuC*	shake flask	glucose	1.7	[75]
*E. coli* DH5α	Δ*nagAB*, Δ*nanATEK*, Δ*ackA*, Δ*poxB*, Δ*ldhA*, (+)*GNA1*, (+)*slr1975*, (+)*glmS**	5 L bioreactor	glucose	7.85	[76]
*E. coli* DH5α	Δ*nagAB*, Δ*nanATEK*, Δ*ackA*, Δ*poxB*, Δ*ldhA*, (+)*GNA1*, (+)*slr1975*, (+)*neuB**, (+)*glmS**; Directed evolution of NeuB was performed using an Neu5Ac aptazyme-based biosensor.	two-stage fermentation	glucose	8.31	[32]
*E. coli* DH5α	Δ*nagAB*, Δ*nanATEK*, Δ*ackA*, Δ*poxB*, Δ*ldhA*, (+)*GNA1*, (+)*neuB*, (+)*glmS**, (+)*age**; Directed evolution of AGE was performed using a Neu5Ac riboswitch-based biosensor.	shake flask	glucose	14.32	[39]
*E. coli* DH5α	Δ*nagEBAC*, Δ*nanATEK*, Δ*ackA*, Δ*poxB*, Δ*ldhA*, Δ*pykA*, Δ*ptsG*, (+)*age*, (+)*neuB*, (+)*GNA1*, (+)*glmS**	shake flask	glucose	16.7	[35]
*E. coli* BL21(DE3)	Δ*nagAB*, Δ*nanATEK*, Δ*manXYZ*, Δ*pykA*, Δ*wecB*, Δ*manA*, (+)*neuB*, (+)*neuC*, (+)*glmU*, (+)*glmM*, (+)*glmS**	3 L bioreactor	glycerol	23.46	[40]
*E. coli* BL21(DE3)	Δ*nagB*, Δ*nanA*, Δ*nanT*, Δ*nanK*, (+)*neuB*, (+)*neuC*, (+)*glmU*, (+)*glmM*, (+)*glmS**	5 L bioreactor	glycerol	46.92	[22]
*E. coli* BL21(DE3)	Δ*nagAB*, Δ*nanATEK*, Δ*manXYZ*, Δ*pykA*, Δ*wecB*, Δ*manA*, Δ*poxB*, Δ*arcA*, Δ*iclR*, Δ*ptsG*, Δ*pfkA*, (+)*neuB*, (+)*neuC*, (+)*glmU*, (+)*glmM*, (+)*glmS**, (+)*nox*, (+)*glf*, (+)*Seppk*; Combining the Neu5Ac TF-based biosensor with HTS by flow cytometry to synergistically optimize the expression levels of *glmS*, *glmM* and *glmU.*	3 L bioreactor	glucose	58.26	[38]
*E. coli* BL21(DE3)	Δ*nanATEK*, Δ*nagAB*, Δ*zwf*, Δ*pfkA*, Δ*ptsG*, Δ*pykA*, Δ*ldhA*, Δ*poxB*, Δ*adhE*, Δ*ackA*, Δ*gldA*, (+)*galP*, (+)*glk*, (+)*glmS**, (+)*GNA1*, (+)*age*, (+)*neuB*, (+)*glmM*, (+)*glmU*, (+)*neuC*, (+)*pck*, (+)*ppsA*, (+)*glpK**	5 L bioreactor	glucose and glycerol	70.4	[55]
*E. coli* W3110	Δ*lacIZ*::P_xylF_-*T7RNAP*, *mlc**, Δ*nagEBAC*, Δ*manXYZ*, Δ*nanATEK*, Δ*ptsG*, Δ*pykA*, Δ*poxB*, Δ*ackA*, Δ*pta*, Δ*iclR*, Δ*rhaB*, (+)*glk*, (+)*glmS*, (+)*yqaB*, (+)*neuB*	5 L bioreactor	glucose	77.12	[27]
*B. subtilis*	Δ*nagAB*, Δ*ldhA*, Δ*pta*, Δ*pyk*, Δ*ptsG*, Δ*gamA*, Δ*gamP*, (+)*glmS*, (+)*GNA1*, (+)*yqaB*, (+)*age*, (+)*neuB*, (+)*pckA*, (+)*ytsJ*	shake flask	glucose and malic acid	2.18	[77]
*B. subtilis*	Δ*nagAB*, Δ*gamA*, Δ*gamP*, Δ*nagP*, Δ*ldhA*, Δ*pta*, Δ*ptsG*, (+)*glmS*, (+)*GNA1*, (+)*yqaB*, (+)*age*, (+)*neuB*, (+)*pyk*, (+)*pfkA*	shake flask	glucose	2.75	[36]
*B. subtilis*	Δ*nagAB*, Δ*ldhA*, Δ*pta*, Δ*pyk*, Δ*ptsG*, Δ*gamA*, Δ*gamP*, (+)*glmS*, (+)*GNA1*, (+)*yqaB*, (+)*age*, (+)*neuB*, (+)*pckA*, (+)*ytsJ*, (+)*nanR*, (+)*folB*	3 L bioreactor	glucose	4.23	[37]
*B. subtilis*	Δ*nagAB*, Δ*ldhA*, Δ*pta*, Δ*ptsG*, Δ*gamA*, Δ*gamP*, Δ*nagP*, (+)*glmS*, (+)*glmU*, (+)*glmM*, (+)*glpK*, (+)*neuB*	3 L bioreactor	glucose and glycerol	21.8	[78]
*B. subtilis*	Δ*nagAB*, Δ*ldhA*, Δ*pta*, Δ*ptsG*, Δ*gamA*, Δ*gamP*, Δ*nagP*, (+)*neuB*, (+)*neuC*, (+)*GNA1*, (+)*age*, (+)*nanE*	5 L bioreactor	glucose	30.10	[21]

Δ: deletion; (+): overexpression; *: mutation.

## Data Availability

No new data were created or analyzed in this study. Date sharing is not applicable.

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
