# Peer review of "Microbial Production of *N*-Acetylneuraminic Acid Using Metabolically Engineered *Escherichia coli* and *Bacillus subtilis*: Advances and Perspectives"

_foods, 2025, doi:10.3390/foods14203478_

Round 1
Reviewer 1 Report
Comments and Suggestions for Authors
Indroduction should include some numbers on market data, amounts produced worldwide and the quantitative use in different applications as well as prices for economical assessment.
Fig.4 contains too much detail, also the resolution is not sufficient for Text and Graphics. It should be split up and allocated in the respective sub-chapters.
Author Response
Comment 1: Introduction should include some numbers on market data, amounts produced worldwide and the quantitative use in different applications as well as prices for economical assessment.
Response: Thank you for your suggestion. In the revised manuscript, we have added relevant market data to the “Introduction” section, including the global market size of sialic acid, and the market demand proportion in different applications (such as medicine, food and cosmetics) (Line 48-52).
Comment 2: Fig.4 contains too much detail, also the resolution is not sufficient for Text and Graphics. It should be split up and allocated in the respective sub-chapters.
Response: Sorry for the low resolution of Figure 4. In the revised manuscript, we have split the original Figure 4 into Figure 4 (revised) (Line 316-326) and Figure 5 (Line 661-665) from the perspective of rational/irrational strategies and adjusted the layout of each part in the figures to make them clearer. Under the irrational strategies, we supplemented the content of optimizing key enzymes (such as AGE and NeuB) through directed evolution (Line 649, 650, Figure 5), making it more comprehensive. Meanwhile, we have reorganized Section 3 to present the strategies for constructing the cell factory for producing Neu5Ac from rational and irrational perspectives. Accordingly, we have made necessary adjustments to the relevant descriptions in the revised manuscript, including the text and Table 2 (Line 22-27, 120-123, 130-134, 296-304, 713-722). To better align with the current content, the titles of Section 3 and its subsections have been updated as follows:
3. Strategies in the construction of cell factories for Neu5Ac production
3.1. Rational metabolic engineering strategies
3.1.1. Relieving feedback regulation
3.1.2. Decreasing the uptake and degradation of Neu5Ac
3.1.3. Reducing the accumulation of by-products
3.1.4. Enhancing the supply of precursors
3.1.4.1. Improving synthetic pathways to enhance ManNAc supply
3.1.4.2. Rewiring the central metabolic module to increase PEP availability
3.1.5. Dynamic regulation
3.1.6. Carbon source optimization
3.1.6.1. Single carbon source
3.1.6.2. Mixed carbon sources
3.2. Irrational strategies

Reviewer 2 Report
Comments and Suggestions for Authors
In general, the manuscript is interesting. I have some comments:
State databases searched, keywords, inclusion/exclusion dates and whether this was restricted to primary research articles or also patents/reviews.
The manuscript notes endotoxin issues with E. coli and the GRAS advantage of B. subtilis but does not quantify downstream implications. Authors should add a subsection on purification challenges (endotoxin removal for E. coli fermentation, chromatography costs, substrate residues from whole-cell catalysis) and, if possible, cite or summarize any available cost or pilot-scale data.
The HTS/biosensor section requires deeper information related to detection ranges and dynamic windows for aptazyme/riboswitch/transcription-factor sensors, false-positive/negative risks, and recommended validation steps post-HTS.
Author Response
Comments 1: State databases searched, keywords, inclusion/exclusion dates and whether this was restricted to primary research articles or also patents/reviews.
Response: The databases we searched mainly included Google Scholar, PubMed and Web of Science; the search keywords included "Neu5Ac", "microbial fermentation" and "metabolic engineering", etc. Additionally, Section 3 exclusively reviewed literature from primary research articles, covering studies conducted from the year 2000 up to the present. Other sections, in addition to reviewing the primary research articles, also referred to relevant reviews.
Comments 2: The manuscript notes endotoxin issues with coli and the GRAS advantage of B. subtilis but does not quantify downstream implications. Authors should add a subsection on purification challenges (endotoxin removal for E. coli fermentation, chromatography costs, substrate residues from whole-cell catalysis) and, if possible, cite or summarize any available cost or pilot-scale data.
Response: Thank you for pointing out this question. In the revised manuscript, we have added Subsection 2.3 entitled "Production methods and purification challenges of Neu5Ac" (Line 271-290), and appropriately supplemented this content in Section 1 and 4 (Line 124-126, 713-716). In this subsection, we discussed the purification challenges in the downstream processing of Neu5Ac production, such as the removal of endotoxins from E. coli fermentation broth, the excessive substrate residues in the whole-cell catalytic synthesis process, and the high cost of chromatography. Additionally, we compared the costs of different Neu5Ac production methods with representative pilot-scale examples in Table 1 (Line 278). To better align with the current content, we updated the title of Section 2 to "Biosynthetic pathway, regulation mechanisms, and production of Neu5Ac".
Comments 3: The HTS/biosensor section requires deeper information related to detection ranges and dynamic windows for aptazyme/riboswitch/transcription-factor sensors, false-positive/negative risks, and recommended validation steps post-HTS.
Response: Thank you for your valuable suggestion. In the revised manuscript, we have further supplemented information on aptazyme, riboswitch, and transcription-factor sensors, including their detection ranges and dynamic windows (Line 672, 673, 676-678). We also noted the risk of false positives and negatives in initial high-throughput screening using biosensors, and that preliminary strains are typically verified through several rounds of well plate or shake flask fermentation (Line 699-702).

Reviewer 3 Report
Comments and Suggestions for Authors
The review summarizes the production of N-acetylneuraminic acid, overall the manuscript is well organized however the corrections provided below can improve the readability and quality of manuscript.
- The manuscripts lack a dedicated section on techno-economic evaluation and scalability strategies
- The regulatory hurdles for Neu5Ac production systems can be provided especially while discussing toxins
- Provide a table with different downstream processing strategies and yields
- Provide future directions for research in this field of work
- Why other microorganisms are not discussed are they not suitable for N-acetylneuraminic acid production?
- Provide a comparative table for coli, B. subtilis, enzymatic, chemical, whole-cell catalysis — with metrics like cost, yield, scalability, safety
- Provide AI/machine learning case studies if available
Author Response
Comments 1: The manuscripts lack a dedicated section on techno-economic evaluation and scalability strategies.
Response 1: Sorry for the previous lack of a dedicated section on techno-economic evaluation and scalability strategies in the manuscript. In the revised manuscript, we have added Subsection 2.3 entitled "Production methods and purification challenges of Neu5Ac" (Line 271-290), and updated the title of Section 2 to "Biosynthetic pathway, regulation mechanisms, and production of Neu5Ac" to make the content more complete. In Subsection 2.3, we evaluated the techno-economic aspects of five production methods. This includes production costs (Table 1), downstream purification costs for endotoxin removal in E. coli fermentation, and for eliminating substrate residues in whole-cell catalysis. Moreover, we also compared the scalability of the above production methods in Table 1, pointing out that the rapid development of relevant technologies has facilitated the large-scale industrial production of Neu5Ac in the text.
Comments 2: The regulatory hurdles for Neu5Ac production systems can be provided especially while discussing toxins.
Response 2: Thank you for your suggestion. In the revised manuscript, we further pointed out the regulatory hurdle for Neu5Ac production systems, mainly including the removal of endotoxins in the fermentation broth of E. coli (Line 281-283).
Comments 3: Provide a table with different downstream processing strategies and yields.
Response 3: Thank you for your suggestion. Based on it, we reviewed the previous downstream processing strategies for the production of Neu5Ac. However, relevant literature reports are scarce, and we are sorry that the information gathered are insufficient to compile a comprehensive table. Therefore, in order to provide an overview of the downstream processing, we have incorporated the downstream purification strategies—primarily chromatography—employed in the production of Neu5Ac through microbial fermentation and whole-cell catalysis, into the revised manuscript (Line 283-290).
Comments 4: Provide future directions for research in this field of work.
Response 4: Thank you for your valuable suggestion. In the revised manuscript, we have pointed out in Section 4 how to achieve both industrial scalability and economic feasibility is the focus of future work. This can be approached from multiple aspects, including optimizing metabolic pathways, improving fermentation processes, and innovating downstream purification technologies, to better promote the development of the Neu5Ac industry (Line 744-748).
Comments 5: Why other microorganisms are not discussed are they not suitable for N-acetylneuraminic acid production?
Response 5: Thank you for highlighting this question. As you mentioned, compared to Escherichia coli and Bacillus subtilis discussed in this review, other microorganisms are currently not the preferred choices as chassis cells for N-acetylneuraminic acid (Neu5Ac) production within the existing literature. Take Corynebacterium glutamicum, a commonly used strain for GlcNAc production via metabolic engineering, as an example. It could be a candidate for Neu5Ac synthesis. However, on the one hand, the naturally existing sialic acid catabolic pathway in it poses an obstacle to the efficient synthesis of Neu5Ac; on the other hand, the long growth cycle of C. glutamicum makes the transition from the growth phase to the production phase more challenging, therefore, it would be more difficult to precisely regulate the balance between the supply of the precursor GlcNAc and PEP. Additionally, some microorganisms that can naturally utilize glucose to produce Neu5Ac, such as E. coli K1, Campylobacter jejuni, and Neisseria meningitidis, can also be considered as candidate microorganisms for the synthesis of Neu5Ac. Yet, their pathogenicity and the associated biosafety risks render them unsuitable for industrial Neu5Ac fermentation. In the revised manuscript, we have supplemented the above reasons for the selection of the chassis utilized in Neu5Ac production (Line 90-97).
Comments 6: Provide a comparative table for E. coli, B. subtilis, enzymatic, chemical, whole-cell catalysis—with metrics like cost, yield, scalability, safety.
Response 6: Thank you for your suggestion. In the revised manuscript, we have provided Table 1 (Line 278), comparing the main production methods of Neu5Ac, including chemical synthesis, enzymatic synthesis, whole-cell catalysis, and microbial fermentation (E. coli, B. subtilis) in terms of cost, yield, scalability, and safety.
Comments 7: Provide AI/machine learning case studies if available.
Response 7: Thank you for your suggestion. In the revised manuscript, we have provided an example of mining Neu5Ac synthase with high catalytic efficiency through the UniKP model by leveraging advanced artificial intelligence techniques and machine learning algorithms (Line 213-221). Following your suggestion, we have conducted a thorough literature review, and subsequently supplemented two latest research advancements on fermentative Neu5Ac production using engineered E. coli in Table 2 (Line 305-307). Accordingly, we have made necessary adjustments to the relevant descriptions in the revised manuscript, mainly concerning the highest titer of Neu5Ac produced through microbial fermentation (Line 85, 86, 198-200, 587-589). In addition, we have supplemented the novel strategies for constructing Neu5Ac cell factories in the above two studies, mainly including: (1) Ma et al. established a coordinated utilization system in E. coli by introducing a new carbon source (glycerol) (Line 613, 614, 630, 631, 633-636); (2) Sun et al. optimized ATP supply by deleting the gene rhaB encoding L-rhamnose kinase, thereby alleviating the problem of by-product acetic acid accumulation (Line 422-427).
Comments 8: The English could be improved to more clearly express the research.
Response 8: Thank you for your suggestion. In the revised manuscript, we have carefully revised the English expressions, focusing on enhancing clarity and readability to better present our research.

Reviewer 4 Report
Comments and Suggestions for Authors
Dear Authors,
The manuscript is well organized and included interesting information that ready for publication.
Best Regards,
Reviewer
Author Response
Comment: The manuscript is well organized and included interesting information that ready for publication.
Response: Thank you very much for your review and approval of this manuscript!

Round 2
Reviewer 3 Report
Comments and Suggestions for Authors
The authors revised the manuscript and can be accepted for publication.